# Challenges and Opportunities in High-Grade Glioma Management and Imaging-Based Response Monitoring During Novel Immunotherapies

**DOI:** 10.3390/cancers17193176

**Published:** 2025-09-30

**Authors:** Carlos A. Gallegos, Benjamin P. Lee, Benjamin B. Kasten, Jack M. Rogers, Carlos E. Cardenas, Jason M. Warram, James M. Markert, Anna G. Sorace

**Affiliations:** 1Department of Biomedical Engineering, University of Alabama at Birmingham, Birmingham, AL 35294, USA; 2Department of Radiology, University of Alabama at Birmingham, Birmingham, AL 35294, USA; 3Medical Scientist Training Program, University of Alabama at Birmingham, Birmingham, AL 35233, USA; 4Department of Otolaryngology, University of Alabama at Birmingham, Birmingham, AL 35233, USA; 5Department of Radiation Oncology, University of Alabama at Birmingham, Birmingham, AL 35233, USA; 6Department of Neurosurgery, University of Alabama at Birmingham, Birmingham, AL 35233, USA; 7O’Neal Comprehensive Cancer Center, University of Alabama at Birmingham, Birmingham, AL 35233, USA

**Keywords:** high-grade glioma, immunotherapy, quantitative MRI, immune-targeted PET, dynamic imaging

## Abstract

Immunotherapy stands as a promising approach to promote durable antitumoral immune responses with potential to achieve long-term remission in high-grade gliomas. The complex inflammatory and immunological mechanisms associated with immunotherapy response can limit the diagnostic utility of conventional imaging approaches, which cannot clearly differentiate tumor progression from treatment-induced effects. Advanced imaging methods are being developed to address these limitations by quantifying key biological features of the tumor microenvironment, providing earlier and more accurate biomarkers of therapeutic response. This literature review summarizes the current imaging strategies used to monitor immunotherapy response in high-grade gliomas, evaluates their advantages and limitations, and highlights advanced modalities with potential to improve early response assessment, guide clinical management, and ultimately improve outcomes.

## 1. Introduction

High-grade glioma (HGG) remains a clinical challenge given its high recurrence rates and limited therapeutic options, resulting in a 5-year survival rate of approximately 10%. Among HGGs, glioblastoma (GBM) is the most prevalent, accounting for over 50% of all malignant CNS tumors, and it is associated with worse clinical outcomes, with a median survival of less than two years [1,2]. To address the poor prognosis of these malignant brain tumors, novel treatment approaches including immunotherapy and multifaceted strategies are under investigation due to their potential to induce durable responses. In this context, the prognostic value of conventional imaging modalities, such as contrast-enhanced *T*_1_- and *T*_2_-weighted magnetic resonance imaging (MRI), is limited by the inability to differentiate tumor progression from therapeutic effects early over the course of treatment. Consequently, novel imaging approaches are being explored in clinical trials and preclinical studies to characterize early immunotherapy-induced effects and predict outcomes in HGG. These efforts aim to identify clinically translatable imaging-based biomarkers for the non-invasive characterization and monitoring of immunotherapy responses, providing clinicians with key tools to predict treatment outcomes to aid in the management of HGG patients. This review highlights current and exploratory treatment strategies for HGG, describes conventional anatomical MRI and positron emission tomography (PET) for diagnosis and response monitoring, and summarizes advanced MRI and PET-based methods with the potential to overcome current limitations in immunotherapy response assessment.

## 2. High-Grade Glioma Classification and Key Biological Characteristics

HGGs encompass the majority of primary malignant brain tumors in adults, with an incidence of approximately 5 cases per 100,000 people, translating to over 20,000 new cases being diagnosed annually in the United States [1]. As implied by their name, high-grade gliomas are classified as the more aggressive higher grades 3 and 4 under the World Health Organization (WHO) Classification of Tumors of the Central Nervous System (CNS) [3,4]. Histopathologically, HGGs are characterized by hypercellularity, nuclear atypia, infiltrative growth and, in the case of GBM, presence of necrosis and microvascular proliferation (Figure 1) [5,6]. In addition to these histological features, HGGs exhibit extensive spatial and temporal intratumoral heterogeneity, denoted by distinct cellular and molecular subpopulations being present within the tumor microenvironment (TME) [7]. Cellular interactions between malignant HGG cells, glioma stem cells (GSC), nonmalignant glial populations, stromal cells, and resident immune cells within this heterogeneous landscape drive tumor progression, invasiveness, and immune evasion [7,8,9]. Furthermore, HGGs can develop functionally and phenotypically distinct tumor regions, or microenvironmental niches, based on the proximity to and eventual absence of normal vasculature as tumors progress [10]. These environmental differences across HGG regions result in the selective promotion of distinct molecular pathways, further enhancing cellular heterogeneity and favoring tumor survival [8,10]. Studies exploring stereotactic image-guided biopsies have further shown distinct molecular profiles, cellular states, and biological signatures across spatially distant samples from a single HGG patient [11]. These findings highlight the limitations of tissue biopsies, as a single sample may fail to capture the inherent spatiotemporal tumor heterogeneity characteristic of HGGs. High intratumoral heterogeneity remains a significant clinical challenge, as therapeutic approaches can be hindered by the diverse and adaptive cell populations within the HGG microenvironment.

## 3. Current Landscape for High-Grade Glioma Treatment

### 3.1. Conventional Therapeutic Approaches for High-Grade Glioma and Their Challenges

The current standard of care for HGG follows a multimodal approach that consists of surgical resection followed by radiation and chemotherapy. As the standard first-line intervention, neurosurgical procedures, ranging from minimally invasive biopsies to craniotomies for gross total resection, are performed to reduce tumor burden and facilitate histological and molecular diagnosis [12,13]. Given the prognostic value of the extent of resection (EOR) in HGG, maximal safe surgical resection aims to maximize tumor removal while preserving neurological function and quality of life [13,14,15,16]. Robotics technologies incorporating image-based neuronavigation are widely utilized in HGG biopsy and standard of care surgical procedures to improve precision [17,18,19,20,21]. Intraoperative imaging during robot-assisted stereotactic procedures has been further demonstrated to enable real-time validation of accurate target site biopsy sampling from tumors while avoiding essential vasculature or eloquent regions [22,23,24,25]. Surgery is standardly followed by concomitant fractionated radiotherapy and temozolomide (TMZ), an alkylating chemotherapeutic agent, based on clinically meaningful findings of survival benefit over radiotherapy alone [26]. In addition, implantation of carmustine wafers, locally placed discs containing alkylating agents, and the use of tumor-treating fields, a noninvasive technique involving low-intensity electric field stimulation, have shown potential for improved survival and are FDA-approved for the treatment of newly diagnosed HGGs [27,28,29]. However, despite these clinical efforts, the majority of HGG patients are expected to exhibit local recurrence and subsequent tumor progression within two years of initial therapeutic intervention, primarily attributed to the infiltrative nature and tumorigenicity of high-grade glioma cells [30,31,32].

Therapeutic regimens for recurrent HGG include a variety of approaches such as secondary resections, TMZ, radiotherapy, molecular therapy and immunotherapy [33]. Secondary surgical resections can help manage symptoms associated with tumor progression and reduce tumor burden, with improved survival observed with an EOR greater than 80% [34,35]. However, this approach can present higher risks of post-surgical complications and neurological sequelae [36]. Repeat radiation has also shown potential in prolonging survival for recurrent HGG patients, with prognostic factors such as age, timing between initial and secondary radiotherapy, and glioma grade influencing outcomes [37]. Furthermore, concurrent radiotherapy and TMZ has shown improved survival benefits compared to other combination systemic therapies [38]. In addition, bevacizumab, a monoclonal antibody specific to the vascular endothelial growth factor (VEGF) that can induce vascular inhibition and normalization, has shown therapeutic benefit in combination with radiotherapy [39,40]. This combination further resulted in decreased rates of radiation necrosis [40], a severe local tissue reaction resultant from radiation-associated neurotoxicity that can limit its therapeutic efficacy [41]. Additional targeted therapies including inhibitors for epidermal growth factor receptor (EGFR), platelet-derived growth factor receptor (PDGFR), and fibroblast growth factor receptor (FGFR) have been explored with limited therapeutic benefit [42,43,44]. Moreover, conventional immunotherapies through immune checkpoint blockade (ICB) therapy targeting the programmed death-1 (PD-1) receptor have been shown to enhance tumor-specific immune activity, leading to significant clinical responses in multiple cancers while having limited efficacy in HGG [45,46,47,48,49]. Nonetheless, prognosis following HGG recurrence remains poor, with therapies offering modest improvements in survival, often increasing it by just a few months [33]. As conventional therapies face limitations in managing HGG, novel treatments that target its heterogeneous and adaptive tumor microenvironment have gained significant attention for their potential to achieve tumor remission and long-term response.

### 3.2. Novel Immunotherapeutic Approaches for High-Grade Gliomas

Novel immunotherapies for HGG include various strategies such as adoptive cell transfer (ACT), cancer vaccines, and oncolytic viruses (OV) [50], see Figure 2. ACT therapy with chimeric antigen receptor (CAR) T cells, which are patient-derived, genetically modified T cells that express synthetic receptors specific to tumor antigens, has shown extensive success in hematological cancers [51,52]. Subsequently, CAR-T cells targeting various tumor-associated antigens, including EGFR variant III (EGFRvIII) and human epidermal growth factor receptor 2 (HER2), are currently being explored in HGG clinical trials [53,54]. Additionally, cancer vaccines utilizing patient-derived dendritic cells (DC) primed with tumor lysates have shown improved survival outcomes in castration-resistant prostate cancer [55], with ongoing clinical trials demonstrating enhanced immune activity and potential for therapeutic benefit in various tumor types including HGGs [56,57,58]. However, implementation of these immunotherapeutic approaches in HGG has been primarily limited by the restrictive nature of the blood–brain barrier (BBB) hindering effective drug delivery, intratumoral heterogeneity allowing for selection of resistant populations and antigen escape, and inherent immunosuppressive TME impairing the antitumoral activity of cytotoxic immune populations [53,59,60,61]. Consequently, novel multifaceted approaches that can successfully address these limitations and promote proper antitumoral immune responses are warranted in HGG immunotherapy.

### 3.3. Oncolytic Herpes Simplex Virus and Its Potential for High-Grade Glioma Treatment

OV therapy has the potential to overcome some of the aforementioned limitations of immunotherapies in HGG. This therapy consists of genetically modified viruses that selectively replicate in and lyse cancer cells while not being pathogenic to normal cells [62]. While intratumoral viral delivery via surgically placed catheters remains the primary approach to circumvent the BBB, early preclinical and clinical evidence has highlighted the potential of intraarterial delivery as a promising less invasive approach for OV tumor delivery [63]. Functionally, virotherapy allows for a multifaceted stimulation of innate and adaptive immune responses through oncolytic release of damage-associated molecular patterns (DAMPs), pathogen-associated molecular patterns (PAMPs), and tumor antigens, promoting a more immunologically activated TME [64]. Furthermore, OVs can be genetically engineered to carry therapeutic genes to be synthesized and released directly within the TME by virally infected cells [65,66,67]. In particular, virotherapy using oncolytic herpes simplex virus (oHSV) type 1 has shown promise for the treatment of HGG due to its effective lytic activity in hypoxic environments and ability to target GSCs, both factors associated with increased tumor aggressiveness and therapeutic resistance [68,69,70]. Clinically, the oHSV-based virotherapy talimogene laherparepvec (T-VEC) has shown therapeutic benefit and durable responses in advanced melanoma and is the first and only approved virotherapy in the United States [65]. Phase I/II clinical trials exploring the use of oHSVs in HGG have further highlighted its potential for improved treatment outcomes while preserving an acceptable safety profile [71,72,73], with G47Δ being approved as a HGG therapeutic in Japan. M032, an oHSV genetically engineered to carry an interleukin (IL)-12 therapeutic payload, has shown enhanced inflammatory responses, immune cell recruitment, prolonged survival, and a similar safety profile when compared to the clinically evaluated G207 oHSV in preclinical studies, supporting its translation into a Phase I/II clinical trial [66]. Given favorable results and preclinical findings of improved virotherapy outcomes in combination with anti-PD1 ICB [74,75], an additional Phase I/II clinical trial exploring M032 and pembrolizumab in recurrent HGG is currently underway (NTC05084430).

## 4. Conventional Imaging Approaches for High-Grade Glioma Response Assessment

### 4.1. Standard of Care Magnetic Resonance Imaging in Immunotherapy Response

MRI serves as the gold standard for diagnosis, treatment planning and response monitoring in neuro-oncology. This modality provides detailed visualization of tissue morphology with improved soft tissue contrast over other imaging approaches such as computed tomography (CT) [76,77]. Conventional anatomical MRI consists of *T*_1_-weighted and *T*_2_-weighted sequences, which rely on differences in tissue relaxation times following a radiofrequency pulse to provide contrast between lipid-rich (e.g., grey and white matter) and fluid-containing (e.g., brain tissue and cerebrospinal fluid) structures, respectively [78]. Radiologically, HGGs are commonly characterized by enhancement in *T*_1_-weighted scans following administration of gadolinium-based contrast agents due to increased BBB permeability from tumor-induced neovascularization and necrosis [6]. However, some reports have shown that a clinically significant subset of histologically confirmed Grade III and IV gliomas (~28%) do not exhibit this radiological property, highlighting the need for more comprehensive non-invasive approaches to characterize gliomas prior to surgery [79]. In addition, these tumors exhibit signal hyperintensity on *T*_2_-weighted and fluid attenuation inversion recovery (FLAIR) sequences, attributed to brain fluid accumulation or edema [80]. Non-invasive assessment of HGGs through these anatomical MRI approaches provides key information for tumor detection, treatment planning, and therapeutic response monitoring.

Multiple guidelines have been utilized to define therapeutic responses in HGG. These have also been adapted to accommodate for emerging therapies and radiological findings. The Macdonald criteria served as one of the earliest guidelines for brain malignancies, which defined progression based on increases (>25%) in the largest cross-sectional area of contrast-enhancing tumor in CT or MRI, or presence of new lesions at least one month apart from the initial scan [81]. Similarly, the response evaluation criteria in solid tumors (RECIST) provided guidelines for defining progression based on a 20% increase in the longest diameter of the target lesion, or appearance of new ones, on CT or MRI [82]. Both of these criteria failed to address key characteristics of brain tumors, including viable non-enhancing components, irregular morphologies, anisotropic growth, and potential for pseudoprogression following chemoradiotherapy or pseudoresponse with antiangiogenic therapies such as bevacizumab [80,83]. Subsequently, the response assessment in neuro-oncology (RANO) group provided updated guidelines which included considerations for non-enhancing tumor regions based on *T*_2_ sequences and an increased window of 12 weeks after chemoradiation for confirmation of radiological progression of >25% increases in perpendicular diameters [84]. Recent updates in 2023 under RANO 2.0 include the change from post-surgical to post-radiotherapy MRI to serve as a baseline scan, in an effort to mitigate the effects of pseudoprogression, and considerations of tumor volume changes as an appropriate assessment approach, with >40% increase being defined as the threshold for progression [85]. Considerations of the heterogeneous and infiltrative nature of the HGG TME as well as therapy-induced changes play a key role in defining radiological guidelines for the assessment and monitoring of clinical responses.

Immunotherapy exhibits radiologically distinct temporal and spatial response patterns that differ from conventional therapeutic approaches. Effective antitumoral immune responses involve both the reinvigoration of inhibited immune cells previously present within the TME and the recruitment of peripheral cells into the tumor, which can take months to develop [86]. Radiologically, this immune cell recruitment and inflammatory responses can result in an apparent tumor enlargement and potential development of new lesions that subsequently stabilize or subside, with similar response patterns observed across multiple cancer types including HGG [87,88]. Therefore, current guidelines for immunotherapy including immune RECIST (iRECIST) for solid tumors, and immune RANO (iRANO) for gliomas recommend continued therapy and follow-up assessment upon early findings of radiographic progression or presence of new lesions [89,90]. Specifically, iRANO allows for a six-month window after the start of immunotherapy in which patients can exhibit radiographic progression as defined by RANO criteria. Within this period, patients with progressive findings should continue treatment with a follow-up assessment no sooner than three months following apparent growth to confirm true tumor progression [90]. This results in a six to nine month window where conventional imaging approaches cannot distinguish true tumor progression from immunotherapy response [91] (Figure 3). Within this timeframe, progressing HGG patients can exhibit significant tumor growth resulting in more limited therapeutic options and worse prognosis. Therefore, novel imaging-based approaches to characterize early immunotherapy response are necessary to provide a wider opportunity window for alternative and more effective therapeutic approaches in HGG.

### 4.2. Conventional Positron Emission Tomography Approaches in High-Grade Glioma Immunotherapy

PET imaging allows for the monitoring of metabolites, proteins, and cellular receptors to inform on key biological processes, relevant for clinical decision making in oncology (Figure 4) [91]. This imaging modality relies on the use of radiotracers, consisting of a targeting moiety labeled with a positron-emitting radioisotope, that selectively accumulate in target regions based on physiological or molecular characteristics. [^18^F]-Fluorodeoxyglucose ([^18^F]FDG) PET, which measures glucose uptake as a surrogate for metabolic activity, is the most widely implemented PET modality for detection, staging, and treatment response assessment in a broad range of malignancies, including breast, colorectal, head and neck, and lung cancers [92]. In brain and CNS malignancies, clinical utility of [^18^F]FDG PET has been limited due to the high metabolic activity of normal brain tissue and activated inflammatory cells, which results in poor target-to-background ratios (TBR) and decreases its ability to distinguish lesions and its prognostic value [91,93,94]. However, some studies evaluating semiquantitative [^18^F]FDG PET metrics, via maximal standardized uptake value (SUV) in the enhancing lesion, have shown potential to predict survival in cases of recurrence treated with chemoradiation and bevacizumab [95,96].

Amino acid PET tracers, such as [^11^C]methionine ([^11^C]MET), [^18^F]fluoroethyl-L-tyrosine ([^18^F]FET) and [^18^F]fluorodihydroxyphenylalanine ([^18^F]FDOPA), show preferential uptake by glioma cells given their increased expression of system L amino acid transporters, allowing for increased contrast in brain lesions [94]. Advanced imaging, especially using amino acid PET, has been shown to aid identification of non-enhancing brain tumor regions that are biologically active or regions within tumors that demonstrate heterogenous response, enabling targeting of these regions for biopsy collection to facilitate accurate diagnosis and treatment planning [97,98,99]. Additionally, these PET tracers have potential to characterize and predict response following chemoradiation and antiangiogenic therapies in HGG, with higher sensitivity than conventional [^18^F]FDG [100,101,102,103,104]. Based on these findings, current PET-based RANO (PET-RANO) guidelines have suggested standardized approaches for assessment of therapeutic response based solely on amino acid PET tracers. These recommendations focus on volumetric quantification of lesions, segmented using an SUV 1.6x greater than background uptake in the frontal lobe, with progression being defined as an average TBR increase of >10%, maximum TBR increase of >30%, or the presence of a new PET-positive lesion at follow-up scans in intervals of 2–3 months following standard of care therapy [94]. Given their value in conventional HGG therapy response, amino acid PET tracers have the potential to inform on positive immunotherapeutic outcomes, which remains a topic of active research [105]. In addition, non-invasive monitoring of tumor hypoxia, via [^18^F]fluoromisonidazole ([^18^F]FMISO), has shown similar potential for predicting tumor progression and survival following chemoradiotherapy, with some evidence for characterizing neuroinflammatory pseudoprogression following ICB in HGG [106,107,108], while remaining investigational as new studies further evaluate its value in response assessment. A summary of the described imaging-based response classification approaches is shown in Table 1.

Despite their potential, PET approaches based on metabolic and microenvironment-targeting tracers can be limited by their inability to directly capture intratumoral immune cell activation and recruitment, which can be prevalent at the early stages of immunotherapy response. Nonetheless, the prognostic value of these PET approaches remains to be explored, with more studies being warranted for proper characterization in the context of HGG immunotherapy.

## 5. Advanced Imaging Approaches for High-Grade Glioma Response Assessment

### 5.1. Biological MRI-Derived Markers and Their Role in Therapeutic Response Assessment

Physiological MRI allows for the non-invasive characterization of biological properties within tissues. In HGG, diffusion-weighted imaging (DWI) and perfusion-weighted imaging (PWI) MRI are collected as part of routine clinical assessment (Figure 5), and there is further validation and reports on how to integrate these into standardized guidelines in neuro-oncology under RANO 2.0 [85,91,109,110]. DWI-MRI involves acquiring multiple images with varying sensitivity to water mobility by applying temporally separated pairs of diffusion-sensitizing gradients with opposite polarities. By acquiring scans with different diffusion weightings (b-values), apparent diffusion coefficient (ADC) maps can be generated to provide information on local tissue diffusion properties [111]. These maps provide quantitative insights into tissue microstructures, as ADC values have been shown to inversely correlate with glioma cellularity based on histological assessments in stereotactic biopsy studies [112,113].

Complementary to this modality, PWI-MRI characterizes tissue vascular properties by evaluating contrast agent kinetics using dynamic susceptibility contrast (DSC) and dynamic contrast-enhanced (DCE) MRI techniques. In brain, DSC-MRI is the most widely adopted method for the assessment and monitoring of tissue perfusion [114]. This modality relies on *T*_2_*-weighted signal decreases induced by the passage of a contrast agent bolus to describe various hemodynamic parameters, including relative cerebral blood volume (rCBV), relative cerebral blood flow (rCBF), and mean transit time (MTT) [115]. Clinically, rCBV is the most studied parameter and has been correlated with microvascular density, vascular proliferation, and angiogenesis in gliomas [116,117,118]. Conversely, DCE-MRI is commonly used for perfusion and permeability imaging in various cancer types and serves as a method to measure BBB disruption [119,120]. This technique evaluates *T*_1_ enhancement from contrast agent leakage into the extravascular extracellular space and can be used to derive contrast kinetic parameters, including transfer constants for influx (*K^trans^*) and efflux (k_ep_), as well as volume fractions for extracellular-extravascular space (v_e_) and intravascular space (v_p_) [119]. Notably, *K^trans^*, k_ep_, and v_e_ have been associated with vascular metrics including vessel size, vascular area, and angiogenesis in gliomas [121]. By providing quantitative information on tumor cellularity and vascularity, physiological MRI techniques have the potential to inform on therapy-induced changes for the monitoring and prediction of therapeutic response in HGG.

Physiological MRI metrics have the potential to characterize early response patterns following both standard of care and novel therapeutic approaches in HGG. Tumor progression is characterized by increased cellular division and angiogenesis, resulting in increased intratumoral vascular and cellular density. These changes translate into decreases in ADC and increases in rCBV and *K^trans^*, which can reflect more malignant phenotypes such as those observed in HGG [122,123]. Following conventional chemoradiotherapy, higher ADC and decreased rCBV and *K^trans^*, indicating decreases in cellular and vascular density, have shown high diagnostic accuracy in differentiating pseudoprogression from true tumor progression [124,125,126]. Furthermore, in the context of bevacizumab, an antiangiogenic therapy, lower rCBV prior to and following treatment were associated with improved survival, while stable ADC has been observed in patients exhibiting pseudoresponse [127,128,129]. Emerging studies assessing physiological MRI response metrics in immunotherapy have highlighted their potential to characterize pseudoprogression and predict survival as previously summarized [130]. In patients treated with ICBs, one study reported that early interval increases in relative ADC (rADC) were associated with treatment response, while another found that solely higher post-treatment rADC correlated with improved outcomes [131,132]. Furthermore, physiological MRI examinations following DC immunotherapy demonstrated that higher minimum baseline rADC was predictive of long-term survival, while decreases in this metric were associated with increased natural killer (NK) cells in peripheral blood [133]. In the same study, changes in maximum lesion rCBV were able to differentiate true progression from pseudoprogression, with similar findings reported by another group [134]. Nonetheless, Song et al. pointed to an absence of prognostic relevance of lesion rCBV during ICB immunotherapy [131], as they found no significant associations of any vascular metric with therapeutic response. When evaluating the findings of these studies, it is important to highlight the inclusion of a subset of patients receiving bevacizumab over the course of ICB in the latter, potentially introducing confounding vascular effects, as well as differences in imaging windows for the post-treatment assessment. The integration of these advanced MRI-based techniques in novel virotherapies for HGG remains an area of growing investigation, as further studies can elucidate on its clinical utility in assessing therapeutic benefit, stratifying patient response, and optimizing therapeutic interventions.

### 5.2. MRI-Based Multiparametric Assessment of Intratumoral Heterogeneity and Its Potential in Immunotherapy Response

Anatomical and physiological MRI approaches in HGG describe distinct tumor characteristics, with each modality offering complementary insights into its morphology and intrinsic biological properties. Computational techniques, such as radiomics and habitat analysis, integrate multiple MRI parameters to enhance tumor characterization by non-invasively evaluating spatiotemporal changes in intratumoral heterogeneity, which is currently limited under conventional biopsy assessment (see Figure 5). Clinical translation of these advanced imaging approaches has the potential to deliver comprehensive biomarkers associated with therapeutic response and survival prediction in HGG.

Radiomics involves the high-throughput extraction of quantitative, multidimensional data, including intensity, texture and semantic features, from radiological images, offering a deeper assessment of tumor pathophysiology [135,136]. In HGG, multimodal radiomic features have shown utility in predicting overall survival prior to standard of care therapy, recognizing pseudoprogression in patients undergoing chemoradiation, and determining markers of progression and survival during treatment with bevacizumab [137,138,139,140,141]. Furthermore, radiomic-based approaches have shown potential to characterize intratumoral immune phenotypes and provide markers for immunotherapy response prediction [142,143]. By extracting key features from clinical images, radiomic-based approaches have demonstrated the potential to improve clinical assessment and provide information for early identification of therapeutic responses. However, their clinical translation has been limited by challenges in reproducibility due to inconsistencies in scanning protocols and feature acquisition techniques [144,145]. Additionally, these features have limited biological interpretability and may fail to capture the microenvironmental heterogeneity of HGGs as they rely on the assumption that tumors are heterogeneous but well-mixed [135,146]. Importantly, there are ongoing efforts aiming to link imaging features with biological pathways via functional radiomics [147], which has potential to increase clinical utility.

Conversely, habitat analysis provides a biologically grounded framework to describe intratumoral heterogeneity by identifying regions with similar biological properties which resemble ecological niches or habitats [148,149]. This approach involves the voxel-wise clustering of parametric MRI maps to define subregions based on shared imaging characteristics. Notably, habitat analysis using physiological cellular and vascular MRI maps has been shown to identify tumor subregions with necrotic, hypoxic, and “enhancing rim” phenotypes that correlate with histological findings in preclinical cancer models [148,150]. In clinical HGG, studies utilizing habitat analysis have shown that increases in hypovascular cellular, or hypoxic, tumor regions are associated with tumor progression, higher recurrence risk and worse therapeutic outcomes following conventional chemoradiotherapy [151,152,153]. This characterization of intratumoral heterogeneity via biologically derived habitats offers a promising approach to assess immunotherapy responses in HGG, as it provides insights into the heterogeneous nature of the TME as well as biological patterns associated with therapy-induced effects and positive prognosis early over the course of treatment.

To characterize imaging features and integrate multimodality data, both radiomics and multiparametric habitat analysis approaches often rely on the implementation of machine learning or artificial intelligence (AI)-based computational methods. These approaches, which consist of computational algorithms capable of identifying patterns within a dataset to discover or predict classifications of new data, have applications at multiple stages of clinical imaging-based assessment. Potential utility of these advanced approaches include image acquisition and reconstruction, tumor segmentation, feature extraction, and development of predictive models [154,155]. In the context of HGG management, AI-based tools have been shown to improve image reconstruction quality, automate the segmentation of tumors, and characterize early therapeutic responses by providing models descriptive of the intrinsic tumor heterogeneity [155,156,157]. Nonetheless, these methodologies require vast and diverse (often multi-institutional) datasets, which can be challenging to collect, and can be influenced by variability in processing workflows and image acquisition parameters. With standardization and collaboration, AI-driven algorithms have the potential to provide unique insights into the heterogeneous TME informative of early immunotherapeutic responses in clinical HGG.

### 5.3. Dynamic PET Imaging to Enhance Tumor Characterization via Mathematical Modeling

Dynamic PET imaging allows for the characterization of tracer perfusion and binding kinetics over time, providing a more comprehensive assessment of underlying biological properties unattainable from standard static acquisitions. This approach involves continuous imaging during tracer circulation, typically over 20 to 60 min based on the expected tracer pharmacokinetics, to generate time–activity curves (TAC) that reflect tracer uptake dynamics within specific tissues [158]. In clinical studies of HGG, semi-quantitative analysis of TACs has shown that lower time-to-peak (TTP) and decreased slope are associated with worse clinical outcomes, therapeutic recurrence, and tumor progression in studies utilizing dynamic [^18^F]FET [101,159,160] and [^18^F]FDOPA [161]. Similar findings were observed in dynamic [^18^F]FET studies in melanoma brain metastasis under ICB immunotherapy, where TTP inversely correlated with pseudoprogression [162]. However, further studies in HGG are required as melanoma brain metastases exhibit differing tumor microenvironment properties with higher immunotherapy response rates [163].

Compartment modeling, a quantitative dynamic PET approach, involves the utilization of biologically based mathematical models to describe tracer pharmacokinetics within and across tissues. This approach typically involves the use of three-tissue compartment models, describing tracer concentrations in blood (C_1_), free tracer within tissue (C_2_), and bound tracer (C_3_). Mathematical modeling of transfer rates across compartments provides quantitative metrics that describe tracer influx (*k*_1_), efflux (*k*_2_), binding (*k*_3_), and unbinding (*k*_4_) [158]. In recurrent HGG patients treated with bevacizumab, combining quantitative compartment modeling metrics from dynamic [^18^F]FET and [^18^F]FDOPA showed greater survival predictive value than standard SUV metrics alone [164]. In addition, preclinical studies have shown that quantitative dynamic PET parameters, including increased [^18^F]FDG binding (*k*_3_) and [^18^F]FET influx (*k*_1_), can distinguish GBM tumors from radiotherapy-induced effects [165]. These quantitative dynamic PET approaches offer significant promise for the kinetic characterization of novel tracers, allowing the assessment of functional changes induced by immunotherapy that can serve as non-invasive biomarkers of early therapeutic response in HGG.

### 5.4. Immune-Targeted PET Imaging of Populations Driving Immunotherapy Response

Immune-targeted PET imaging allows for the non-invasive, longitudinal monitoring of biomarkers associated with immune cells and their function, providing key spatial and temporal information for the assessment of immunotherapy response. Among immune cell populations, T cells, characterized by cluster of differentiation 3 (CD3) expression, play an essential role in antitumoral immune responses due to their ability to recognize tumor antigens, trigger apoptotic pathways in cancer cells, and provide long-term immunological memory [166]. Preclinical immune-targeted PET imaging using a ^89^Zr-labeled anti-CD3 antibody showed that increased intratumoral T cell infiltration correlated with subsequent tumor volume reduction in colon cancer models treated with ICB therapy [167]. Furthermore, helper and cytotoxic T cell subpopulations expressing CD4 and CD8, respectively, have also been explored as potential markers of immunotherapy response. CD4+ helper T cells contribute to antitumor immunity by supporting the differentiation of cytotoxic T cells, producing effector cytokines such as interferon γ, and, under specific conditions, inducing direct cytotoxicity on tumor cells [168,169]. Immune-targeted PET using a CD4-targeting antibody fragment successfully stratified immunotherapy responders in multiple syngeneic mouse tumor models [170]. However, expression of CD3 and CD4 on T regulatory cells (T_regs_), a subset of helper T cells associated with immunosuppressive activity including secretion of anti-inflammatory cytokines and mediated cytotoxic activity on effector immune cells [169], can limit the prognostic value of these targeted immune-targeted PET approaches.

Conversely, CD8-targeted PET imaging offers a promising alternative as it directly reflects the distribution and kinetics of cytotoxic effector T cells, key mediators of antitumoral activity and immunotherapy efficacy. Similar to other T cell markers, CD8-targeted PET has demonstrated utility in characterizing immune dynamics and predicting immunotherapy response in preclinical cancer models [171,172,173]. This immune-targeted PET approach has further been shown to allow for the monitoring and visualization of CD8+ cell populations in HGG models [174,175], with some studies highlighting its prognostic value for immunotherapy response [176]. Moreover, antibody-derived CD8-tracers, including [^89^Zr]Zr-ZED88082A and [^89^Zr]Zr-IAB22M2C, have been evaluated in Phase I and I/II clinical trials. These studies demonstrated safety and provided an insight into the spatiotemporal dynamics of CD8+ T cell populations, with the potential to predict immunotherapeutic responses [177,178,179]. While HGG patients were not evaluated on these trials, higher uptake of [^89^Zr]Zr-ZED88082 was reported in a brain metastatic lesion compared to healthy brain in a melanoma patient. This may indicate its potential use for brain lesions, though differences in tumor microenvironments in HGG would warrant further exploration [179]. Complementary to this approach, PET imaging of granzyme B, a serine protease secreted by activated cytotoxic immune cells, using radiolabeled peptides provides a functional marker of immune-mediated tumor killing and correlates with immunotherapy responses [180,181,182]. This non-invasive strategy to assess effector immune cell function is currently in Phase I clinical trials (NTC04169321, NTC05888532). See Figure 6 for representative schematic of advanced PET-based approaches. Non-invasive assessment of cytotoxic populations and their effector molecules provides a promising approach to characterize mechanisms that result in positive immunotherapy responses, offering potential markers for patient stratification and therapy optimization in HGG.

## 6. Current Outlook and Future Perspectives

As novel therapeutic approaches for HGG are increasingly being translated from preclinical to clinical settings, it is critical to develop and validate imaging techniques that can overcome the limitations of conventional anatomical assessments and allow for the early identification of treatment responses. In this context, it becomes essential to acknowledge the underlying biological characteristics that influence radiological findings, particularly in treatment strategies that significantly alter physiological features or cellular trafficking. This has been highlighted as a main factor that limits the prognostic value of anatomical *T*_1_-weighted MRI, where BBB permeability can be altered by transient inflammatory, vascular, and cellular effects and can result in signal hyperintensity increases not associated with true tumor progression. Immunotherapy presents additional considerations, as treatment response relies on the promotion, reinvigoration, and recruitment of immune cell populations. These present unique spatiotemporal characteristics that differ from direct cancer cell cytotoxicity observed during standard-of-care treatment or molecular and targeted therapies. These differences can play key roles when evaluating advanced MRI and PET-based approaches for the prognostic assessment of novel immunotherapies.

As part of routine clinical assessment for HGG, DWI and PWI MRI have demonstrated prognostic value for the early characterization of immunotherapy outcomes. Decreased cellular density, denoted by increased normalized ADC, has been consistently found to be descriptive of response across various retrospective HGG studies evaluating different immunotherapies [131,132,133]. While immune cell infiltration could result in transient increases in cellular density, these studies provide evidence for decreased cellularity as an early predictive biomarker of response, with findings as early as one month following the start of therapy [132]. Nonetheless, consensus remains to be reached on the appropriate quantitative metric, as pre-treatment, post-treatment, and interval changes in rADC were independently found to be descriptive of response. Conversely, the value of vascular assessment, defined by rCBV, remains inconclusive as clinical immunotherapy studies have reached different conclusions, potentially influenced by confounding factors following multiple lines of treatment when evaluating heterogeneous patient populations. As vascular accessibility can influence the ability of systemic therapies to reach the tumor microenvironment and further provide pathways for immune cell infiltration, imaging techniques targeting perfusion could have unexplored prognostic relevance for the early assessment of immunotherapy response. Future dedicated imaging studies that support these findings for each independent immunotherapy are essential to properly define the prognostic value of these quantitative MRI approaches and pursue their integration into clinical immunotherapy guidelines for HGG. During these studies, prospective efforts to standardize time windows for pre- and post-treatment imaging, patient inclusion criteria, acquisition protocols, and computational analysis workflows are necessary to mitigate confounding factors and promote the translatability across clinical and academic centers.

Molecular monitoring via amino acid PET imaging has demonstrated significant value during diagnosis, treatment planning, and response monitoring in neuro-oncology, leading to the definition of standardized guidelines and recommendations for its use despite not being integrated into standard of care imaging [94,183,184]. As an imaging approach that targets metabolic activity, amino acid PET offers unique advantages over conventional contrast-enhanced MRI, as signal hyperintensities are informative of intrinsic tumor biological characteristics while having decreased influence from BBB disruption effects. Particularly for HGG, studies have highlighted the potential of amino acid PET for delineation of metabolically active tumor regions outside *T*_1_ and *T*_2_ enhancing sites [185,186]. With the development and clinical integration of simultaneous PET/MR systems, novel multimodal strategies evaluating quantitative MRI-based cellular and vascular metrics in PET-defined biological tumor volumes could yield essential information for early therapeutic response assessment and longitudinal monitoring of local and satellite tumor recurrence. During immunotherapy, multiple studies, summarized in the literature [187], have highlighted the competitive nature of amino acid uptake between tumor and immune cells, providing a potential connection between decreased intratumoral amino acid PET tracer uptake and improved immunological activity as a metric for early response assessment. Future studies integrating amino acid PET imaging during immunotherapy in HGG are therefore warranted given their potential to improve therapeutic response assessment and complementary nature with standard of care imaging methods.

Monitoring of immune populations via immune-targeted PET remains primarily investigational through preclinical studies and early-stage clinical trials. In HGG, preclinical studies by Hunger et al. and our group have focused on the monitoring of cytotoxic immune populations and demonstrated the association of positive immunotherapy outcomes with spatial homogeneity in intratumoral T cell distribution [176,188]. In this context, it is important to highlight that these modalities target components of the TME that can be completely or partially absent, particularly prior to therapeutic interventions. Therefore, their clinical use should be paired with a modality that allows for tumor segmentation, such as anatomical MRI or amino acid PET, in a post-immunotherapy setting where immune cell recruitment can be expected. For clinical translation, the tracer’s ability to cross the BBB should also be considered, as that factor could limit its accessibility to tumor regions where BBB remains intact, while integration of dynamic kinetic models could help evaluate and predict tracer uptake within these regions. Similar to other PET-based approaches, additional preclinical and clinical studies are needed to properly characterize the prognostic value of immune-target PET imaging and define optimal imaging windows to capture the key spatiotemporal immune kinetics that drive immunotherapy response in HGG.

Advanced computational approaches including radiomics and habitat analysis allow for the description and longitudinal monitoring of tumor heterogeneity by extracting imaging features beyond conventional quantification. These approaches can provide metrics descriptive of tumor immunological pathways and properties, as well as delineate biologically distinct intratumoral sites, with the potential to improve early diagnostic and prognostic accuracy without the need to acquire complementary imaging sequences or modalities. For translation into clinical practice, it is essential to pursue consistency across acquisition and analysis protocols as the resulting metrics heavily rely on imaging features susceptible to signal variation across images. In particular for AI-based methods, data harmonization and expansion of multicenter datasets will be fundamental to improve the robustness and clinical integration of these tools. Current efforts for the standardization of image acquisition parameters, radiomic feature extraction, and biological interpretability of imaging features can aid in the development of robust generalizable algorithms to support the clinical management of HGG [147,189,190]. As emerging technologies improve computational processing times and integrate novel algorithms with intuitive image visualization tools, future studies can further explore and validate the clinical utility of these analysis tools for the monitoring and characterization of response to immunotherapy during HGG clinical trials.

In addition to the need for harmonization across institutional image acquisition protocols, cost and accessibility remain barriers to the advancement of novel imaging approaches into standard of care. Translation of these imaging modalities to mainstream clinical use will require demonstrated cost-effectiveness, combined with measurable clinical benefit, to ensure widespread utilization [191]. Further, due in part to high procurement and operational costs associated with advanced imaging, geographic and systemic barriers exist for implementation, especially for patients in rural areas or with limited access to large research institutions [192,193]. Finally, regulatory approval and insurance reimbursement of novel imaging agents or quantitative image analysis may slow clinical translation [194,195]. Despite these challenges, implementation of advanced imaging techniques for both diagnostic and therapeutic use continues to grow and remains a promising approach for treatment planning and response monitoring in HGG.

## 7. Conclusions

Non-invasive characterization of intratumoral biology and its heterogeneity from a spatial tissue perspective offers unique approaches to understand the spatiotemporal changes induced by novel therapeutic approaches. Integration of these advanced imaging methods into clinical workflows can support early response stratification, allowing for a wider window to pursue alternative therapies in patients with limited response and avoiding unnecessary, and often more aggressive, interventions in patients who show favorable responses. Ultimately, these imaging-guided strategies have the potential to allow for personalized treatment strategies, optimizing therapeutic outcomes and improving clinical management for HGG patients.

## Figures and Tables

**Figure 1 cancers-17-03176-f001:**
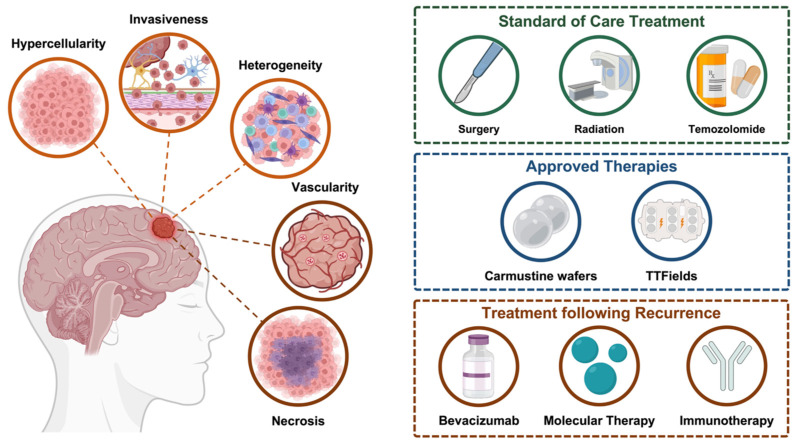
High-grade glioma is characterized by hypercellularity, invasive growth and high cellular heterogeneity. Grade 4 gliomas, such as glioblastoma, can further exhibit microvascular proliferation and necrosis. Standard of care therapeutic approaches consist of maximal safe surgical resection followed by concurrent fractionated radiation and temozolomide, with carmustine wafers and tumor treating fields (TTFields) being approved for newly diagnosed high-grade gliomas. Upon recurrence, therapeutic options consist of repeat resection, irradiation, temozolomide, bevacizumab, targeted therapies, and immunotherapy.

**Figure 2 cancers-17-03176-f002:**
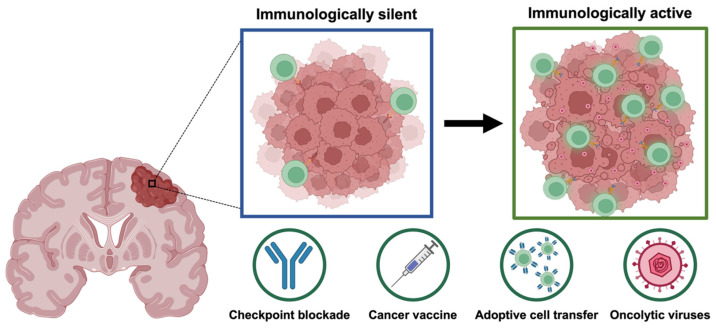
Immunotherapy aims to promote an immunologically active tumor microenvironment by enhancing immune infiltration and localized inflammatory pathways. Immunotherapies explored in high-grade glioma include checkpoint blockade inhibitors, cancer vaccines, adoptive cell transfer, and oncolytic viruses.

**Figure 3 cancers-17-03176-f003:**
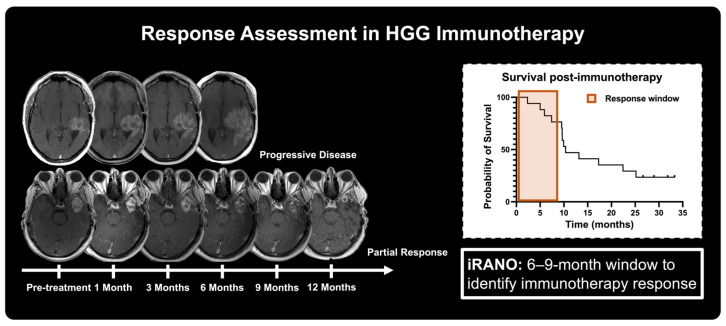
Treatment response monitoring in high-grade glioma relies primarily on anatomical post-contrast *T*_1_-weighted and *T*_2_-weighted sequences collected prior to and following treatment administration. Monitoring of changes in lesion dimensions of MRI scans plays a key role in the determination of response with considerations for tumor grade and therapeutic approach. Representative schematic shows response classification for HGG following oHSV immunotherapy as determined by iRANO, where a six-month window for response following signs of tumor progression is given due to the potential influence of inflammatory responses in radiological findings.

**Figure 4 cancers-17-03176-f004:**
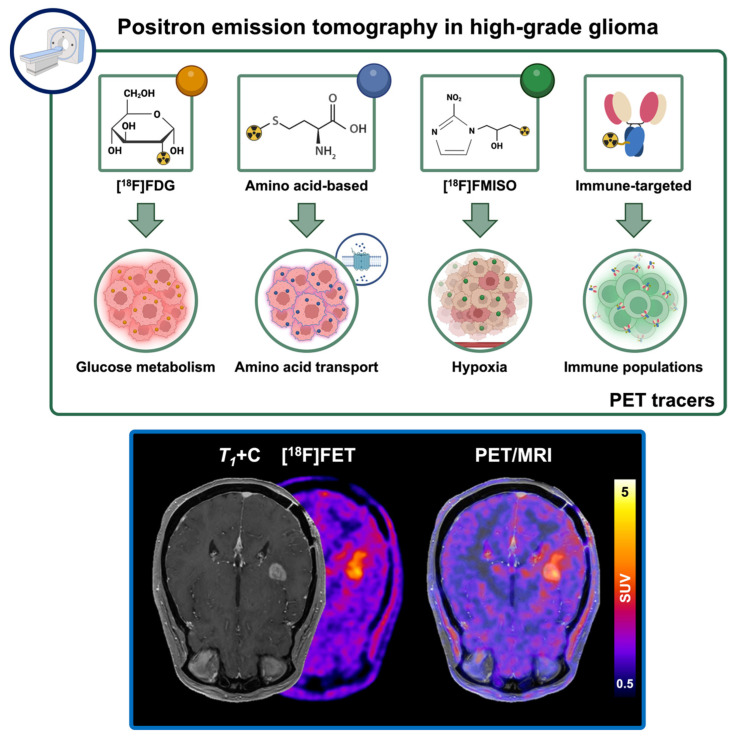
Positron emission tomography allows for the non-invasive characterization of metabolic and functional mechanisms through the use of radiotracers including [^18^F]FDG, [^18^F]FMISO, and amino acid-based and immune-targeting tracers. Combination of PET and MRI modalities combines the specificity of PET with the high anatomical resolution of MRI, providing essential information for the non-invasive characterization of lesions in neuro-oncology.

**Figure 5 cancers-17-03176-f005:**
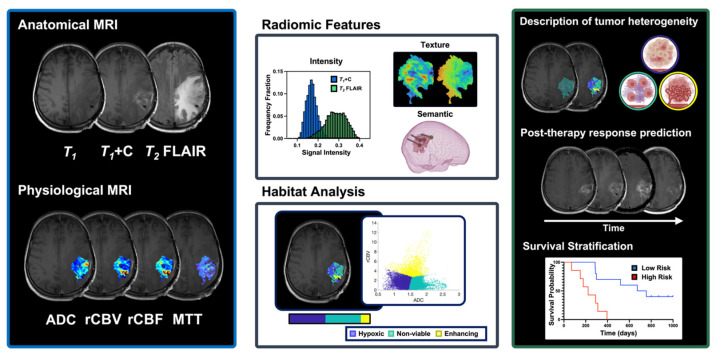
Advanced quantitative MRI modalities such as diffusion- and perfusion-weighted MRI provide key biological information related to tumor cellularity and vascularity, with key roles in the assessment and monitoring of HGG prior to and following therapeutic interventions. Furthermore, voxel-wise information from these MRI maps allows for the collection of radiomic features, including intensity, texture, and semantic metrics, and definition of biologically distinct intratumoral habitats. These approaches provide essential information on intratumoral heterogeneity for the early prediction of therapeutic responses and stratification of patient survival in HGG.

**Figure 6 cancers-17-03176-f006:**
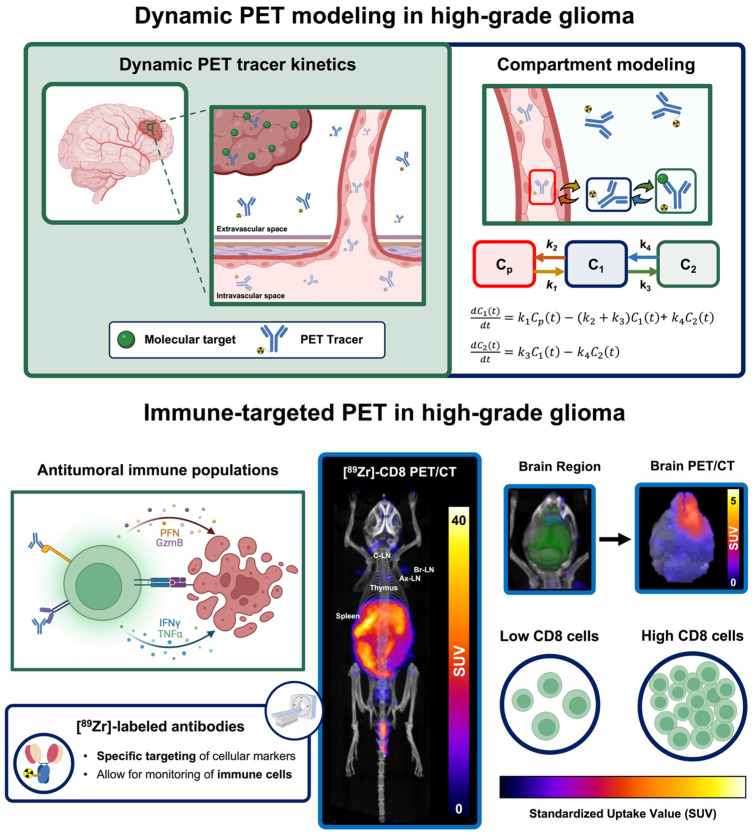
Dynamic PET imaging enables characterization of tracer pharmacokinetics through serial acquisition of tracer localization. Mathematical modeling methods allow for the definition of differential equations that can be fitted to collected PET data to obtain quantitative metrics associated with tracer influx (*k*_1_), efflux (*k*_2_), binding (*k*_3_), and unbinding (*k*_4_) rates informative of tumor biological properties. In addition, antibody-based immune-targeted PET tracers allow for the targeting of immune populations based on biomarkers such as CD3, CD4, and CD8 to inform on immune cell infiltration and characterize immunotherapy-induced mechanism associated with treatment response in immunotherapy. Representative schematic shows CD8-targeted PET imaging of a murine GBM model following treatment with combination oHSV and ICB immunotherapy (adapted from [176]).

**Table 1 cancers-17-03176-t001:** Summary of criteria for disease progression in high-grade gliomas.

Criteria	Modality	Metric	Criteria for Progressive Disease (PD)
Metric Change	New Lesions	Clinical Decline	*T*_2_ Worsening
RECIST [82]	CT, *T*_1_ + C MRI	Longest diameter	>20% increase	Considered PD	Not evaluated	Not evaluated
McDonald [81]	CT, *T*_1_ + C MRI	Largest cross-sectional area	>25% increase	Considered PD	Considered PD	Not evaluated
RANO [84]	*T*_1_ + C, *T*_2_ MRI	Sum of bipendicular diameters	>25% increase	Considered PD	Considered PD	Considered PD
RANO 2.0 [85]	*T*_1_ + C, *T*_2_ MRI	Sum of bipendicular diameters	>25% increase	Considered PD	Considered PD	Considered PD
Volume	>40% increase
iRANO [90]	*T*_1_ + C, *T*_2_ MRI	Sum of bipendicular diameters	>25% increase	Not considered PD	Considered PD	Not considered PD
		TBR_max_ of lesion	>30% increase			
PET-RANO [94]	Amino acid PET	TBR_mean_ of lesion	>10% increase	Considered PD	Considered PD	N/A
		PET-positive volume	>40% increase

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
