# Peer review of "Challenges and Opportunities in High-Grade Glioma Management and Imaging-Based Response Monitoring During Novel Immunotherapies"

_cancers, 2025, doi:10.3390/cancers17193176_

Round 1
Reviewer 1 Report
Comments and Suggestions for Authors
The manuscript is titled “Challenges and opportunities in high-grade glioma management and imaging-based response monitoring during novel im-3 monotherapies. The authors present a complete and well-written review that discusses the limitations of conventional imaging techniques, such as MRI and PET, in distinguishing true tumor progression from treatment-induced effects in HGG, and then explore advanced imaging methods, including quantitative MRI and immune-targeted PET, highlighting their potential to provide earlier and more accurate biomarkers for therapeutic response. Further explains how these advanced techniques can characterize key biological features of the tumor microenvironment, offering improved tools for clinical management and patient outcome prediction in the context of emerging immunotherapies like oncolytic viruses and cell transfer therapies. In conclusion, the article is a strong and informative review.
Please try to address the points below to strengthen the articles.
- Authors should discuss clinical translation barriers such as cost, reproducibility, regulatory approval, and standardization of acquisition protocols.
- To strengthen the review article, authors in the manuscript should discuss how artificial intelligence and robotics technologies are specifically utilized to advance imaging or computational imaging.
- Discussion of the practical challenges related to, or implicitly connected with, recent AI-driven advancements in computational imaging?
Reviewer 2 Report
Comments and Suggestions for Authors
The text by Gallegos C., et al. deals with a review on therapeutic results on malignant gliomas. The text is synthetic and interesting; however, the authors should focus on some points:
The title indicates two terms which are contradictory a review, “challenges and opportunities in …” indicate a selection of topics that does not reflect the academic goals of a review; I think that these terms, which are clearly, and in my opinion, adequately discussed in the text (L523-620) should be deleted from the title.
- Similarly, other terms used should be clarified (e.g. “addressed” on L21; L24-26).
- Advantages of imaging studies are clearly and briefly explained on figures 3-6. Also, highlights of immunotherapy are well illustrated on figures 1 and 2.
